# Ruminal Microbiota and Fermentation in Response to Dietary Protein and Energy Levels in Weaned Lambs

**DOI:** 10.3390/ani10010109

**Published:** 2020-01-09

**Authors:** Xiaokang Lv, Kai Cui, Minli Qi, Shiqin Wang, Qiyu Diao, Naifeng Zhang

**Affiliations:** Feed Research Institute of Chinese Academy of Agricultural Sciences, Key Laboratory of Feed Biotechnology of the Ministry of Agriculture and Rural Affairs, Beijing 100081, China; 13121191399@163.com (X.L.); cuikai@caas.cn (K.C.); diaoqiyu@caas.cn (Q.D.)

**Keywords:** lambs, growth performance, microbial diversity, rumen fermentation, rumen morphology

## Abstract

**Simple Summary:**

Ruminants, such as sheep, are economically important because they contribute to digesting and converting plant materials into edible meat and milk for humans to consume. An adequate plane of nutrients, such as energy and protein, is essential for rumen development and growth. However, sheep production is mostly affected by inadequate nutrition in rural areas of China. As one of the most prolific and perennial estrus breeds in China, Hu sheep has huge potential for catering to the growing meat demands of the market and consumers. In this study, the effects of dietary energy and protein levels on growth performance, microbial diversity, and physiological properties of the rumen in weaned lambs were evaluated. The results showed that a low dietary energy level restrained growth performance and changed the microbiota and associated ruminal fermentation phenotypes of lambs. However, protein had a minor effect. The findings are of great significance for promoting rumen development and establishing the optimal nutrient supply strategy for lambs.

**Abstract:**

Supplying sufficient nutrients, such as dietary energy and protein, has a great effect on the growth and rumen development of ruminants. This study was conducted to evaluate the effects of dietary energy and protein levels on growth performance, microbial diversity, and structural and physiological properties of the rumen in weaned lambs. A total of 64 two-month-old Hu lambs were randomly allotted to 2 × 2 factorial arrangements with four replicates and with four lambs (half male and half female) in each replicate. The first factor was two levels of dietary metabolizable energy (ME) density (ME = 10.9 MJ/Kg or 8.6 MJ/Kg), and the second factor was two levels of dietary crude protein (CP) content (CP = 15.7% or 11.8%). The trial lasted for 60 days. A low dietary energy level restrained the growth performance of lambs (*p* < 0.05). The ruminal concentration of acetate and the ratio of acetate to propionate increased but the propionate concentration decreased significantly with the low energy diet. However, the rumen morphology was not affected by the diet energy and protein levels. Moreover, a low energy diet increased ruminal bacterial diversity but reduced the abundance of the phylum Proteobacteria (*p* < 0.05) and genus *Succinivibrionaceae*_uncultured (*p* < 0.05), which was associated with the change in ruminal fermentation phenotypes. By indicator species analysis, we found three indicator OTUs in the high energy group (*Succinivibrionaceae*_uncultured, *Veillonellaceae*_unclassified and *Veillonellaceae*_uncultured (*p* < 0.01)) and two indicator OTUs in the low energy group (*Bacteroidales*_norank and *Lachnospiraceae*_uncultured (*p* < 0.01)). In conclusion, these findings added new dimensions to our understanding of the diet effect on rumen microbial community and fermentation response, and are of great significance for establishing the optimal nutrient supply strategy for lambs.

## 1. Introduction

Ruminants, such as sheep, are economically important because they contribute to digesting and converting plant materials into edible meat and milk for humans to consume. Rumen development is critical for the utilization of solid feed in ruminants [1]. Rumen development includes microbial colonization [2], anatomic development [3], and functional achievement [4]. The dry-feed intake stimulates microbial proliferation and volatile fatty acid (VFA) production in the rumen, which reflects the capacity of the ruminal microbiota to degrade feedstuffs [5]. An adequate plane of nutrition is essential for rumen development of lambs. The main focus is to supply sufficient dietary energy and protein, as these nutrients greatly affect growth and rumen development [6]. Dietary energy also affects the utilization of other nutrients, such as protein [7,8]. A complex inter-relationship exists between dietary nutrients, rumen microbial composition, fermentation function, and morphology. Mullins et al. found that rumen weight is significantly affected by dietary energy levels, and a high energy diet increased rumen weight [9]. The rumen fermentation parameters, including pH, volatile fatty acids, and ammonia nitrogen, are influenced by dietary protein levels in goats and cows [10,11]. Therefore, the rumen fermentation environment is closely related to dietary nutrients such as energy and protein. Nevertheless, little is known about the effects of dietary energy and protein levels on rumen microbiota development and its regulatory effect on the fermentation and anatomic maturation of rumen in Hu sheep lambs.

The sheep industry is one of the key parts of the supply-side structural reform of agriculture in the new era of China. It is vital to the revitalization and eradication of poverty for farmers from remote areas and ethnic minorities. As one of the most prolific and perennial estrus breeds, Hu sheep has huge potential for catering to the growing meat demands of the market and consumers [12]. Prolific Hu sheep can supply more lambs for fattening, especially in the intensive systems of China. The better the lambs are reared, the more fattening sheep and breeding ewes could be supplied for the industry. Sheep production is affected mostly by inadequate nutrition in rural areas. As a unique Chinese sheep breed [13], the differences in growth rates and reproductivities indicate differences in their nutritional requirements. Whether dietary energy and protein levels have an interactive effect on rumen fermentation and microbial composition in lambs remains unclear in Hu sheep lambs. There must be some connection between rumen microorganisms, rumen fermentation, and evolution of the rumen morphology [1,2]. Therefore, we hypothesized that diets with different energy and protein levels could change ruminal microbiota and fermentation, thereafter affecting the development of rumen epithelial papillae. The objective of this study was to explore the impact of dietary energy and protein levels on the microbial diversity and the structural and physiological properties of the rumen in weaned lambs. In this study, 16S rDNA sequencing was used to investigate the effect of different protein and energy levels on ruminal microbiota to gain a preliminary understanding needed for further research aimed at improving the feed utilization efficiency of Chinese Hu sheep.

## 2. Materials and Methods 

### 2.1. Animals, Diets and Experimental Design

This research was conducted at the Hailun sheep industry Co., Ltd., Jiangsu, China (latitude 32.30′ N, longitude 119.54′ E). The experimental procedure was approved by the Chinese Academy of Agricultural Sciences Animal Ethics Committee (AEC-CAAS-FRI-CAAS20180602), and humane animal care and handling procedures were implemented throughout the experiment.

Sixty-four 60 d old Chinese Hu sheep lambs with an average body weight of 14.95 ± 0.56 kg were used for a 60 d feeding trial. Lambs were reared in one shed on a commercial farm. Two male and two female lambs were assigned to one pen based on body weight; in this way, the experiment consisted of 4 treatments with four pens per treatment. The 16 pens used in the experiment were 9 m^2^ (3 m × 3 m) with a bamboo leaky floor, automatic waterers, and 3 m fence-line feed bunks. During the experimental period, lambs were twice fed a pelleted starter (diameter, 4 mm; length, 10 mm) at 08:00 and 16:00 h in a 50:50 proportion (as a feed basis). Water and starters were provided ad libitum during the whole experiment period.

Experimental lambs were randomly allotted to 2 × 2 factorial arrangement. The first factor was two levels of dietary metabolizable energy (ME) density (ME = 10.9 MJ/Kg or 8.6 MJ/Kg), and the second factor was two levels of dietary crude protein (CP) content (CP = 15.7% or 11.8%). Hence, there were four treatments with four replicates, using the pen as an experimental unit.

Diets were formulated according to the nutrient requirements of small ruminants (2007) and the Chinese Feeding Standard of sheep (NY/T 816-2004). The high energy and high protein diets were targeted for lambs between 15 and 30 kg body weight with the average daily gain of 200 g/d. Then, 20% of the energy and protein contents were deducted to form the low energy and/or protein diets. The other nutrients of the experimental diets were met or exceeded the nutrient requirements of lambs. The ingredients and chemical composition of the pelleted starter are presented in Table 1. The amount of daily feed offered and refused were recorded. The lambs were weighed before the morning meal on 60, 90, and 120 days. The dry matter intake (DMI), average daily gain (ADG), and ratio of feed to gain (F/G) were calculated accordingly.

### 2.2. Feed Samples Chemical Analyses

The amount of feed offered to each pen and residuals were collected and weighed daily for determining feed intake; nearly 500 g of feed samples from each pen were collected for chemical analysis. Feed samples were dried in a forced-air oven at 65 °C for 48 h and then stored in sealed plastic containers at 4 °C until analysis for dry matter (DM), organic matter (OM), crude protein (CP), ether extract (EE), gross energy, calcium, phosphorus, neutral detergent fiber (NDF), and acid detergent fiber (ADF) according to the Association of Official Analytical Chemists [14].

### 2.3. Rumen Sample Collection

According to the health condition and body weight of the lambs, one healthy lamb with a body weight close to the average of the group was selected from each pen at the age of 120 days. In this way, 16 lambs with 4 lambs per treatment (male and female in half) were selected and slaughtered. After slaughter, the rumen was separated and opened. Rumen digesta was collected and its pH was measured immediately after collection using a digital pH meter (PB-10; Sartorius, Goettingen, Germany). Thereafter, approximately 50 mL of the digesta was filtered through four layers of gauze. A 10 mL sample of the strained fluid was collected, acidified with 2 mL 25% (w/v) metaphosphoric acid, and frozen at −20 °C for analysis of volatile fatty acids (VFA) and ammonia nitrogen (NH3-N) [15]. A 2 mL Rumen content sample (mixed liquid and solid contents) was transferred into a plastic tube and stored at −80 °C for analysis of ruminal microbiota. The weight of the rumen, reticulum, omasum, and abomasum was measured after cleaning and eliminating the contents. For rumen morphological development, rumen tissue specimens (~2 cm × 2 cm) were obtained from the ventral sac and fixed in 4% formaldehyde. 

### 2.4. Measurement of Rumen Morphology

The rumen tissue samples were dehydrated with an ethanol and toluene (Beijing Chemical Works) series and embedded in paraffin (Leica, Wetzlar, Germany). After fixation, tissue specimens were trimmed and processed according to standard histological procedures, and then they were stained with hematoxylin and eosin. Three pieces of the disconnected section of each sample were observed, and the papillae length, papillae width, and muscular layer thickness were measured using the Image-Pro express image analysis and processing system. Morphometric analyses were performed at a magnification of 4 × 10 times (Olympus BX-51; Olympus Corporation, Tokyo, Japan) using Image-Pro Plus 6.0 (Media Cybernetics, Silver Spring, MD, USA).

### 2.5. Ruminal Fermentation Parameters

The rumen fluid samples were thawed at 4 °C, and the concentration of ammonia nitrogen was determined by the phenol-sodium hypochlorite colorimetric method. VFA concentration was determined by gas chromatography (GC) using methyl valerate as the internal standard in an Agilent 6890 series GC equipped with a capillary column (HP-FFAP19095F-123, 30 m, 0.53 mm diameter, and 1 cm thickness). Samples were injected using an auto-sampler (AI 3000, Thermo Scientific, Waltham, MA, USA) into an AE-FFAP capillary column (30 m × 0.25 mm × 0.33 μm, ATEO, Lanzhou, China) on a Varian GC (TRACE 1300, Thermo Scientific, MA, USA). Samples were run at a split ratio of 20:1 with a column temperature of 45 °C to 150 °C with an increase of 10 °C/min followed by a 5 min hold. The injector and detector temperatures were 200 °C and 250 °C, respectively. Peak integration was performed using Chromeleon^®^ 7 Software (Thermo Fisher Scientific Inc., Waltham, MA, USA). All ruminal fluid samples were assayed in duplicate.

### 2.6. Rumen Microbial Diversity Data

Microbial DNA was extracted from the rumen content using a commercial DNA Kit (Omega Bio-tek, Norcross, GA, USA) according to the manufacturer’s instructions. Amplification and sequencing were performed as described elsewhere [16]. The V3–V4 region of the bacterial 16S ribosomal RNA genes was amplified using primers 338F (5’-barcode-ACTCCTRCGGGAGGCAGCAG)-3’ and 806R (5’-GGACTACCVGGGTATCTAAT-3’), where the barcode is an eight-base sequence unique to each sample. 

Amplicons were purified using the AxyPrep DNA Gel Extraction Kit (Axygen Biosciences, Union City, CA, USA) according to the manufacturer’s instructions and quantified using QuantiFluor™ -ST (Promega, Waltham, MA, USA). Purified amplicons were pooled in equimolar and paired-end sequenced (2 × 250) on an Illumina MiSeq platform (Illumina Inc., San Diego, CA, USA) according to the standard protocols.

### 2.7. Statistical and Bioinformatics Analysis

Raw sequences were filtered through a quality control pipeline using the Quantitative Insight into Microbial Ecology (QIIME v 1.9.1, the Knight and Caporaso labs, Flagstaff, AZ, USA) tool kit [17]. The denoised sequence was clustered into operational taxonomic units (OTUs) using UPARSE according to the sequence identity of 97% [18]. The phylogenetic affiliation of the 16S rRNA gene sequence was analyzed against the SILVA (SSU115) 16S rRNA database using Ribosomal Database Project (RDP) Classifier (http://rdp.cme.msu.edu/) with a confidence threshold of 70% [19,20]. Richness estimators Chao1 and observed OTUs, as well as rarefaction curves were calculated for the overall bacterial community using QIIME. Rarefaction curves were constructed in R using mothur rarefaction analysis [21]. The sequencing data of this research was submitted to the Sequence Read Archive (SRA) with an accession number of SRP239068

The interactive analysis among bacteria and metabolic phenotypes was performed in the R package (V3.1.0, Robert Gentleman and Ross Ihaka, Auckland, NZL). Spearman’s rank correlations and *p*-values were calculated and plotted using the packages ‘hmisc’ and ‘corrplot’. 

Differences in growth performance, rumen morphology, rumen fermentation parameters, and the rumen microbial diversity between the four groups were analyzed using the SAS (v 9.4, 2004, SAS Institute, Inc., Cary, NC, USA ) general linear model (GLM), using the pen as an experimental unit. All data were analyzed using a model that included the fixed effects of dietary energy level, dietary protein level, and the interaction between energy and protein levels, as well as the random effect of each pen.

The following model was fitted to the data:Y_ijk_ = μ + α_i_ + β_j_ + α_i_ × β_j_ + γ_k_ + e_ijk_,(1)
where Y_ijk_ is the dependent variable, μ is the overall mean, α_i_ is the fixed effect of dietary energy level, β_i_ is the fixed effect of dietary protein level, α_i_ × β_j_ is the fixed effect of the interaction between energy and protein levels, γ_k_ is the random effect of pen, and e_ijk_ is the random residual error.

Duncan’s multiple range test was used to identify differences between specific treatments when a significant difference existed. A *p*-value < 0.05 was considered to indicate a statistically significant difference.

Indicator species analysis was performed using the multipat function of the indicspecies package in R (version 3.6.0) [22]. The *p*-values were corrected for multiple comparisons using the false discovery rate (FDR) with the Benjamini–Hochberg method.

## 3. Results

### 3.1. Growth Performance

The effects of dietary protein and energy levels on growth performance of lambs are shown in Table 2. There were no interactions (*p* > 0.05) between dietary energy and protein levels for the growth performance of lambs. Nevertheless, a low dietary energy decreased (*p* < 0.05) the final body weight, ADG, and enhanced (*p* < 0.05) the feed to gain ratio of lambs compared with high energy diets (Table 2). However, lambs consuming diets containing different protein contents had no significant changes (*p* > 0.05) in the DM intake, ADG, and the ratio of feed to gain. 

### 3.2. Rumen Fermentation and Morphology

Likewise, there were also no interactions (*p* > 0.05) between dietary energy and protein levels for the rumen fermentation and morphology of lambs (Table 3). Low dietary energy significantly increased (*p* < 0.05) the percentage of acetate and isobutyrate but decreased (*p* < 0.05) the percentage of propionate and valerate in rumen fluid. Consequently, lambs fed a low energy diet had a greater ratio of acetate to propionate in comparison with those fed a high energy diet (*p* < 0.05). Correspondingly, a decreased rumen weight and its ratio to complex stomach were observed in lambs fed a low dietary energy diet (*p* < 0.1). The rumen pH, NH_3_-N, total VFA concentrations, papillae length, papillae width, mucosal thickness, and base thickness were not affected by the dietary energy and protein levels (*p* > 0.05). 

### 3.3. Sequences and Alpha Diversity

The sequencing of the amplicon libraries resulted in a total of 704,849 raw reads prior to quality checking and assigning the reads to their respective samples. After applying quality control, Miseq sequencing of the V3–V4 region of the 16S rDNA gene in all the rumen fluid samples resulted in a total of 575,435 clean sequences being identified, of which the average length was 440.99 bp, and 72.6% of the sequence length was between 441 and 460 bp. Based on the 97% sequence identity, 575,435 bacterial sequences were assigned to 3894 OTUs. The results of good coverage showed that 99%–100% of the microbial species were sampled for the groups of microorganisms, indicating that the sequencing effort had sufficient sequence coverage for each microbial group.

Diversity concerns both taxon richness and evenness, and our results demonstrated that both parameters were increased in low energy groups compared with high energy groups (Table 4). The observed OTU, Chao1, Ace, Shannon, and Simpson indices supported the order described above, and there were significant differences between the high and low energy groups (*p* < 0.05). However, dietary protein levels had no significant effect on the alpha diversity of lambs (*p* < 0.05). Moreover, no interactions were observed between dietary energy and protein levels on the alpha diversity of lambs (*p* < 0.05).

### 3.4. Taxonomy of Rumen Bacterial Composition

Sixteen different phyla were detected in these samples, and seven of them were abundant over 1% of the total sequences at least in one group (Figure 1A). On average of the four groups, the Bacteroidetes were the most dominant, accounting for 44.51% of total sequences, followed by bacteria from the Firmicutes (28.87%), Proteobacteria (20.78%), and Spirochaetes (2.79%). No interaction was observed between dietary energy and protein levels for the relative abundance of bacteria at the phyla level, except Bacteroidetes. With the low energy diets, the relative abundance of Bacteroidetes with high protein levels were 59.8% higher than low protein diets, while with the high energy diets, the relative abundance of Bacteroidetes did not differ between protein levels (Table 5, Figure 2A). Among the 16 phyla, the relative abundance of Proteobacteria decreased, whereas that of the Bacteroidetes, Firmicutes, and Lentisphaerae increased in low energy diets compared with high energy diets (*p* < 0.05). No effect was observed of protein level on the abundance of phyla bacteria, except Bacteroidetes, which was significantly increased (*p* < 0.05).

One hundred and twenty-six different genera were detected in these samples, and thirty of them were abundant over 1% of the total sequences at least in one group (Figure 1B). On average, the most abundant genus was *Prevotella*, accounting for 18.82% of total sequences (Table 5), followed by bacteria from *Succinivibrionaceae_uncultured* (16.26%) and *RC9_gut_group* (11.32%). There was an interaction effect (*p* < 0.05) between dietary energy and protein levels on the relative abundance of *BS11_gut_group_norank* (Figure 2B), *S24-7_norank* (Figure 2C), and *Ruminococcus* (Figure 2D). In the low energy diets, the relative abundance of *BS11_gut_group_norank* with high protein levels were higher (*p* < 0.05) than low protein diets, whereas the relative abundance of *S24.7_norank* and *Ruminococcus* with high protein levels were lower (*p* < 0.05) than low protein diets. In contrast, in high energy diets there were no differences (*p* > 0.05) in the relative abundance of *BS11_gut_group_norank*, *S24.7_norank* and *Ruminococcus* between protein levels. Lambs fed a low energy diet had a higher abundance of *Succinivibrionaceae_uncultured* but had a lower abundance of *Bacteroidales_norank*, *Lachnospiraceae_uncultured*, *Pseudobutyrivibrio*, *Ruminococcaceae_uncultured*, *Saccharofermentans*, and *RFP12_gut_group_norank* than the high energy diet group (*p* < 0.05). 

### 3.5. Indicator Genera Analysis

To further ascertain which genus is responsible for the observed community differentiation between HE and LE, we used indicator analyses to discover significant associations between genera and treatments. Indicator genera and their corresponding indicator values can be found in Table 6. When we excluded genera with a relative abundance of less than 1%, we found three indicator genera in the HE groups (*Succinivibrionaceae_uncultured*, *Veillonellaceae_unclassified* and *Veillonellaceae_uncultured* (*p* < 0.01)), and two indicator genera in the LE group (*Bacteroidales_norank* and *Lachnospiraceae_uncultured* (*p* < 0.01)).

### 3.6. The Relationship Between Rumen Microbiota and Metabolic Phenotypes

Spearman’s correlation coefficient (r) was used to determine correlations between rumen fermentation parameters and any of the genus that was abundant over 1% at least in one group (Figure 3). The abundance of bacteria at the genus level and the rumen fermentation parameters were considered to be correlated with each other if the correlation coefficients were above 0.57 and the *p*-value was below 0.05. The propionate proportion was positively correlated with bacteria, such as *Succinivibrionaceae_uncultured*, *Selenomonas*, *Veillonellaceae_uncultured, Veillonellaceae_unclassified, and Succinivibrio*, but negatively correlated with bacteria such as *Saccharofermentans*, *Bacteroidales_norank*, and *Ruminococcaceae_uncultured*. The acetate proportion was negatively correlated with *Succinivibrionaceae_uncultured, Veillonellaceae_uncultured, Veillonellaceae_unclassified,* and *Selenomonas,* but positively correlated with bacteria such as *Ruminococcus, Ruminococcaceae_uncultured, RF16_norank,* and *SHA.109_norank*. Conversely, isobutyrate and isovalerate proportion were negatively correlated with bacteria such as *Succinivibrionaceae_uncultured* but positively correlated with bacteria such as *Ruminococcaceae_uncultured*, and *Saccharofermentans*. A:P was positively correlated with bacteria, such as *Ruminococcaceae_uncultured* and *Saccharofermentans*, but negatively correlated with bacteria such as *Succinivibrionaceae_uncultured* and *Selenomonas*. Butyrate proportion and NH_3_-N concentrations were not significantly correlated with many genera.

## 4. Discussion

The growth of animals is largely dependent on the amount of dry matter intake and the dietary levels of protein and energy [23]. To optimize sheep production, emphasis on their energy and protein requirements and their rumen health, especially the ruminal microbiota and its relationship with the metabolic phenotypes, was essentially needed. Lowered energy and protein levels may result in a shortage of fermentable ME and nitrogen for rumen microorganisms and, thereafter, a depression in the synthesis of microbial protein and in the amount of protein available to the animal, which may consequently decrease the weight gain. Sayed found that decreasing the dietary digestible energy from 14.7 to 12.2 MJ/kg resulted in a reduced average daily gain and an increased daily feed intake in lambs fed a diet containing 14.7% protein [24]. Abbasi et al. also found that lowered dietary energy level (9.1 vs. 10.7 MJ ME/kg) reduced the average daily gain and increased the feed conversion ratio in Kamori Goat Kids [25]. Our previous study [1] focused on 20–60-day-old Hu sheep lambs also found that low energy or protein diet restrained growth performance. Furthermore, this study strengthened that low energy (10.9 vs. 8.6 MJ ME/kg) diets reduced the ADG of lambs and enhanced the ratio of feed to gain. As both the current and previous studies confirmed, a low energy ration has negative effects on the growth performance of lambs. However, the effects of protein levels on the growth of lambs are inconsistent. Some reports indicated that protein levels play an important role in DM intake and ADG in lambs, especially the dietary protein levels below 17% [26]. Nevertheless, Kaya et al. observed no differences in the DMI, ADG, or GF when 13% and 16% protein levels were compared in Morkaraman lambs in Turkey [27]. Similarly, the protein level did not affect the ADG or ratio of feed to gain of Hu sheep lambs in this study when 11.8% to 15.7% protein levels were compared. It is thought that when the protein supply exceeds the requirement, energy becomes a limiting factor for growth, and the animals no longer respond to additional intakes of protein. The growth of animals have been reported to vary according to the species of animal, environment, and trait [28]. Hu sheep is a slow-growing, local breed traditionally raised under backyard systems in the rural areas of China. It has lower productive performance (daily weight gain and feed conversion) compared to those modern commercial meat-type breeds. It also has been suggested that Hu sheep may use non-conventional feed resources, fibrous feed with low energy, and protein content. Although information on the nutrient requirement of Hu sheep is lacking, probably the CP requirement of Hu sheep lambs could be reduced to 11.8% without a detrimental effect on growth performance during this growth stage. Besides, the no-growth response of lambs for the protein level should be associated with the unchanged DMI in this study, for the dietary protein concentration affects the growth rate by affecting the dry matter intake of lambs [26]. Therefore, the dietary energy levels may play a more important role than protein levels in growth performance of Hu sheep lambs.

Optimal development of rumen is required to ensure animal health and productivity. Sun et al. [10] and Cui et al. [1] found that lowered dietary protein or energy levels significantly reduced rumen weight. However, under their study, the ratio of rumen to complex stomach was not affected by the level of dietary energy or protein. As obtained in this study, dietary protein levels had no significant effect on the rumen weight, ratio of rumen to complex stomach, rumen papilla height, width, base thickness, or mucosal thickness. Nevertheless, the lowered energy level had a trend to decrease the rumen weight and its ratio to complex stomach. The reason for this phenomenon may be that the rumen morphology of lambs tends to mature at this stage and is less affected by dietary energy and protein levels.

In this study, the ratio of NFC to NDF was slashed by nearly 70% with low energy diets compared with high energy diets. It means a vast change of fermentable carbohydrate and metabolizable energy. Higher NDF content would encourage the proliferation of cellulolytic bacteria and change the fermentation model of rumen microorganisms. Consisted with previous studies [6,23], the percentage of acetate and the ratio of acetate to propionate increased, and the percentage of propionate reduced under low energy in this study. These changes mean that energy level altered the rumen fermentation patterns. Yang et al. reported that with an increase in diet digestible carbohydrates (starch, 32.9 vs. 24.1), the propionate concentration of the rumen fluid also increased [29]. It was confirmed that dietary energy levels have a significant effect on the rumen fermentation environment by regulating the microbiota of the rumen. Ruminal NH_3_-N concentration is a crude predictor of the efficiency of dietary degradable nitrogen conversion into microbial nitrogen [30]. However, there was no significant effect of dietary protein and energy levels on the concentration of NH_3_-N in rumen fluid, although it was numerically over 60% higher in high energy and high protein diets compared with other diets. Statistically, the dietary protein level might have fulfilled or exceeded the requirements of lambs, and thereby has a minor effect on the rumen fermentation.

In this study, MiSeq sequencing of 16S rRNA was used to assess the effects of different dietary energy and protein levels on microbial community structure in lambs. We found that the microbial diversity of lambs fed low energy diets was higher than those fed high energy diets. Other researchers reported that bacterial diversity is affected by diet, such as the dietary concentrate to forage ratio [2,31]. Qian et al. [31] and Tapio et al. [2] found the high forage to concentrate ratio increased the diversity and richness of ruminal bacteria, confirmed by a high Firmicutes at the phylum level and *Prevotella* at the genus level. Our previous study [1] also found that a low energy diet increased OTUs and Bacteroidetes abundance in the rumen of lambs. The cellulolytic bacteria are relatively abundant, and similar observations were found in the gut of humans on higher fiber diets [1], suggesting that their metabolic role may be important in a low energy or high NDF diet [32]. The composition of the low energy diet that contains a high fiber content has been explained as the main cause [31]. 

Low energy diets resulted in significant shifts in the structure of the rumen microbial community as compared with the high energy diets. In the present study, the phyla Bacteroidetes, Proteobacteria, and Firmicutes were the dominant bacteria among the four groups. It is consistent with a previous study conducted by Rey et al. [33]. A low dietary energy level significantly increased the abundance of Firmicutes, Bacteroidetes, and Lentisphaerae, and decreased the abundance of Proteobacteria. Firmicutes mainly decompose fibrous substances [34]. The diet NDF level of the low energy group is higher than that of high energy group. That is why lowering the energy level of the diet enhanced the abundance of the Firmicutes in this experiment. Pitta et al. reported that there was an increase in the number of Bacteroidetes within the rumen in buffalo fed high levels of hay, which decreased the dietary energy concentration [35]. Bacteroidetes are believed to produce degradation enzymes, targeting plant cell wall compounds (e.g., cellulose and pectin) and leads to the release of VFA (mainly acetate, propionate, and butyrate) [36]. Nevertheless, Bacteroidetes are still responsible for protein fermentation [37]. These results suggested that the altered bacterial community composition linked with the production of different VFAs. The present study revealed that the genera *Prevotella*, *RC9_gut_group*, *BS11_gut_group_norank*, and *Succinivibrionaceae*_*uncultured* and *Ruminococcaceae*_*uncultured* dominated in the four groups. These results were consistent with the conclusions of other studies [38,39]. Low dietary energy levels significantly decreased the abundance of *Succinivibrionaceae_uncultured* but increased the genera *BS_11_gut_group_norank, Ruminococcaceae_uncultured, Ruminococcus,* and *Lachnospiraceae_uncultured*. Mccabe et al. [40] found that restriction of feed resulted in a large reduction of an uncharacterized *Succinivibrionaceae* species (OTU-S3004)*. Succinivibrionaceae_uncultured*, as well as OTU-S3004, belong to the family Succinivibrionaceae (phylum Proteobacteria), which was suggested to generate propionate by the succinate pathway. Therefore, the substantial reduction of Proteobacteria might be the caution for the decreases of propionate and increased ratio of A:P in the low energy diets in this study. The genus *BS_11_gut_gorup_norank* belongs to family BS_11_gut_group (phylum Bacteroidetes). Solden et al. indicated that *BS11* is specialized to ferment many different hemicellulosic monomers (xylose, fucose, mannose, and rhamnose), producing acetate and butyrate for the host [37]. The genus *Ruminococcaceae_uncultured* and *Ruminococcus* belong to the family Ruminococcaceae (phylum Firmicutes). As a member of VFA producers, Ruminococcaceae has been well illustrated to be responsible for the degradation of diverse polysaccharides and fibers [41]. Therefore, it may be presumed that high NDF content in the low energy level benefited the growth and predominance of Ruminococcaceae and the increase of acetate content in the rumen fluid. The genus *Lachnospiraceae_uncultured* belongs to family Lachnospiraceae (phylum Firmicutes). This family contains many known plant-degraders and most of the butyrate-producers in the gut [42]. Therefore, Firmicutes may be the functional core rumen phylum, performing essential functions in energy conversion. *Prevotella*, belonging to the phylum Bacteroidetes, is the most abundant genus in the adult rumen and is thought to account for starch utilization, protein degradation, and peptide absorption and fermentation [43]. However, the effect of dietary nutrients on the abundance of *Prevotella* is inconsistent. Petri et al. reported that an increased ratio of concentrate-to-forage improved the abundance of *Prevotella* in the rumen of cattle [44]. Jami et al. argued that the content of *Prevotella* was unaffected by the diet structure [45]. The mechanism that the genus *Prevotella* was not influenced by dietary nutrition in this study is not clear yet, and further studies are required.

It is extremely important to know which species affects the classification of high-energy and low-energy microbes. We used indicator species analysis to identify species that were statistically significant indicators of different groups. At the genus level, dominant members of the HE group samples mainly contained *Veillonellaceae_unclassified*, *Succinivibrionaceae_uncultured*, and *Veillonellaceae_uncultured*. LE groups were dominated by *Bacteroidales_norank*, and *Lachnospiraceae_uncultured*. The Veillonellaceae family includes members that could produce propionate as their major fermentation product [46]. Creevey et al. found that the bacteria isolated from *Succinivibrionaceae_uncultured* are associated with starch degradation and propionate production [47]. Therefore, we believe that *Veillonellaceae_unclassified, Succinivibrionaceae_uncultured* and *Veillonellaceae_uncultured* are indicator species of the high energy group. 

The diet-induced changes in rumen microbial community structure are related to changes in ruminal fermentation parameters. The present study revealed the interplay patterns of rumen microbiota and metabolic phenotypes in the rumen of lambs fed different energy level diets. Correlation analysis showed that the concentration of acetate, isobutyrate, and isovalerate was negatively correlated with *Succinivibrionaceae_uncultured* but positively correlated with *Ruminococcaceae_uncultured*. Whereas the concentration of propionate and valerate were positively correlated with *Succinivibrionaceae_uncultured* but negatively correlated with *Ruminococcaceae_uncultured*. *Succinivibrionaceae_uncultured* was significantly decreased in low energy diets. Therefore, the findings indicated that *Succinivibrionaceae_uncultured* is associated with the energy level of diets [40] and is a diet-specific genus. It has been reported that butyrate can promote the growth of rumen papillae [48]. The content of butyrate in the rumen fluid is related to *Butyrivibrio* [49]. Nevertheless, there was no significant difference in the abundance of *Butyrivibrio* between the four treatment groups in this study, which is most likely the reason for no difference in butyrate concentration. This also explains why there is no difference in rumen papillae length between the four treatments. 

## 5. Conclusions

Low energy diets significantly decreased the growth performance and the ruminal propionate proportions, but increased acetate proportion and the ratio of acetate to propionate through expanding the diversity of ruminal bacteria and shaping the ruminal microbiota structure of weaned lambs. Ruminal microbiota is more sensitive to dietary energy than to dietary protein in weaned lambs. Dietary energy content drives phenotype variation of lambs, and this study solidifies the notion that microbial phenotype is also influenced by dietary nutrients. The rumen bacteria profiles of lambs fed high-energy and -protein levels diet was dominated primarily by Bacteroidetes, Proteobacteria, and Firmicutes, and in particular, representatives of the genera such as *Prevotella*, *RC9_gut_group*, *Succinivibrionaceae_uncultured*, and *Veillonellaceae_uncultured*. The abundance of genera *Succinivibrionaceae_uncultured* (phylum Proteobacteria) is associated with the change in ruminal fermentation phenotypes. These findings are of great significance for promoting rumen development and establishing the optimal nutrient supply strategy for lambs.

## Figures and Tables

**Figure 1 animals-10-00109-f001:**
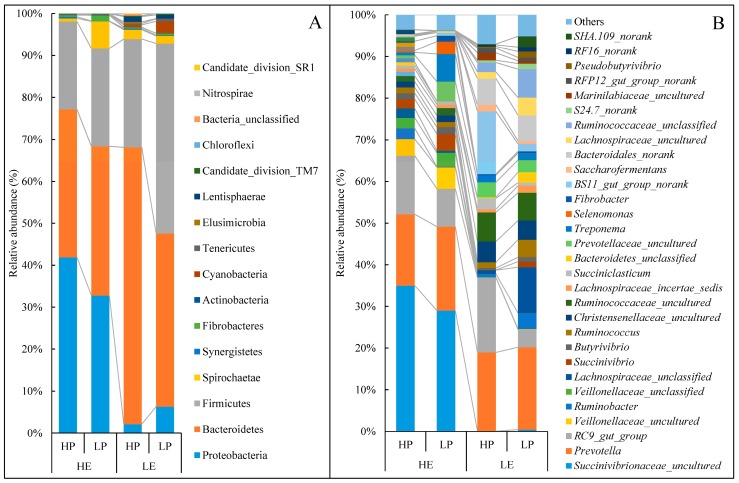
Effects of dietary energy and protein levels on phylum-level (**A**) and genera-level (**B**) composition of the rumen microbiota, represented as average abundance per diet. HE, high energy level, ME = 10.9 MJ/Kg; LE, low energy level, ME = 8.6 MJ/Kg; HP, high protein level, CP = 15.7%; LP, low protein level, CP = 11.8%.

**Figure 2 animals-10-00109-f002:**
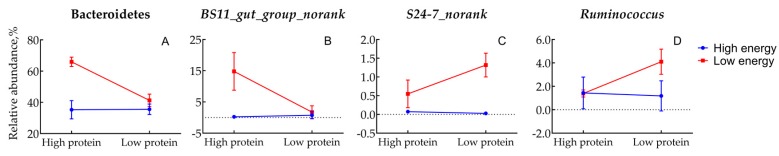
Interaction effects between dietary energy and protein levels on the relative abundance of Bacteroidetes (**A**), *BS11_gut_group_norank* (**B**), *S24-7_norank* (**C**), and *Ruminococcus* (**D**). High energy level, ME = 10.9 MJ/Kg; low energy level, ME = 8.6 MJ/Kg; high protein level, CP = 15.7%; low protein level, CP = 11.8%.

**Figure 3 animals-10-00109-f003:**
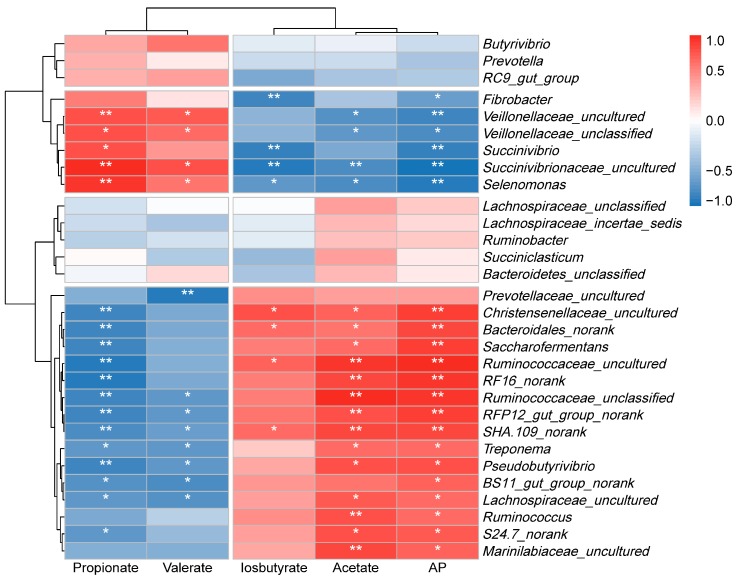
Relationships between bacterial communities and fermentation parameters in the rumen of lambs fed different protein and energy level diets. Strong correlations are indicated by red and blue colors with 1 indicating a perfect positive correlation (dark red) and −1 indicating a negative correlation (dark blue), whereas weak correlations are indicated by white colors. Spearman test, * *p* < 0.05, ** *p* < 0.01.

**Table 1 animals-10-00109-t001:** Ingredients and chemical composition of experimental starters.

Items	HE^5^		LE^5^	
	HP^5^	LP^5^	HP	LP
Ingredients, %				
Maize	49.3	62	20	31.1
Wheat bran	4.4	0	16.5	14.5
Soybean meal	7.3	0	20.5	11.3
Alfalfa meal	35	31	0	0
Straw meal	0	3	38	38
Limestone	0	0	1	1.1
Premix ^1^	4	4	4	4
Total	100	100	100	100
Nutrients ^2^, % of DM				
DM ^3^	87.17	86.96	88.62	88.34
ME, MJ/kg	10.92	10.92	8.64	8.64
CP	15.74	11.78	15.72	11.82
RDP	8.75	5.96	7.47	5.27
EE	3.38	3.42	1.97	2.13
NDF	23.35	22.20	45.71	44.49
ADF	14.36	13.73	25.86	25.00
NFC ^4^	47.04	52.85	26.91	32.48
NFC/NDF	2.01	2.38	0.59	0.73
RDP/ME, g/MJ	8.01	5.46	8.65	6.10
Ca	1.09	0.99	0.92	0.93
TP	0.60	0.53	0.63	0.59

^1^ The premix provided the following per kilogram of the diet: VA 12,000 IU, VD 2000 IU, VE 30 IU, Cu 12 mg, Fe 64 mg, Mn 56 mg, Zn 60 mg, I 1.2 mg, Se 0.4 mg, Co 0.4 mg, Ca 3.2 g, P 1.2 g, and NaCl 6.4 g; ^2^ Nutrients are all measured values except ME; ^3^ The ME of the starter was calculated according to Tables of Feed Composition and Nutritive Values in China 2012 and Feeding Standard of Sheep (NY/T 816-2004); ^4^ NFC = 100 − NDF − CP − EE − Ash; DM—dry matter; ME—metabolizable energy; CP—crude protein; RDP—rumen degradable protein; EE—ether extract; NDF—neutral detergent fiber; ADF—acid detergent fiber; NFC—non-fiber carbohydrate; Ca—calcium; TP—total phosphorus. ^5^ HE—high energy level, ME = 10.9 MJ/Kg; LE—low energy level, ME = 8.6 MJ/Kg; HP—high protein level, CP = 15.7%; LP—low protein level, CP = 11.8%.

**Table 2 animals-10-00109-t002:** Effects of dietary energy and protein levels on the growth performance of lambs.

Items	HE^1^		LE^1^		SEM	*p*-Value ^2^		
	HP ^1^	LP ^1^	HP	LP		E	P	E × P
Initial body weight, kg	14.95	14.84	14.99	15.02	0.26	0.636	0.870	0.773
Final body weight, kg	29.90 ^a^	29.46 ^a^	27.51 ^b^	27.81 ^b^	0.58	0.003	0.891	0.488
Average daily gain, g/d	249.24 ^a^	243.63 ^a^	208.53 ^b^	213.05 ^b^	6.73	0.000	0.930	0.421
Dry matter intake, g/d	1235.17	1259.16	1260.10	1275.03	15.98	0.184	0.204	0.759
Ratio of feed to gain	4.97 ^b^	5.18 ^b^	6.05 ^a^	6.00 ^a^	1.65	0.000	0.602	0.410

^a,b,c^ Different superscript letters in the same variable indicate statistical differences (*p* < 0.05). SEM = Standard error of the mean; ^1^ HE, high energy level, ME = 10.9 MJ/Kg; LE, low energy level, ME = 8.6 MJ/Kg; HP, high protein level, CP = 15.7%; LP, low protein level, CP = 11.8%. ^2^ E = energy effect; P = protein effect; E × P = interaction effect of energy and protein.

**Table 3 animals-10-00109-t003:** Effects of dietary energy and protein levels on rumen fermentation of lambs.

Parameter	HE ^2^		LE ^2^		SEM	*p*-Value ^3^		
	HP^2^	LP^2^	HP	LP		E	P	E × P
**Fermentation Parameters**								
pH	7.05	6.94	7.28	7.26	0.08	0.115	0.695	0.786
NH_3_-N, mmol/L	19.02	10.60	9.00	11.76	1.99	0.276	0.479	0.174
TVFA, mmol/L	21.63	29.21	18.80	16.10	2.71	0.163	0.657	0.356
Acetate, %	64.39 ^bc^	62.70 ^c^	69.22 ^ab^	70.10 ^a^	1.10	0.003	0.812	0.452
Propionate, %	21.78 ^ab^	26.50 ^a^	13.67 ^c^	15.34 ^bc^	1.74	0.002	0.227	0.554
Butyrate, %	4.69	4.69	5.49	4.89	0.24	0.343	0.567	0.558
Valerate, %	0.79 ^a^	0.83 ^a^	0.41 ^b^	0.39 ^b^	0.08	0.010	0.956	0.841
Isobutyrate, %	3.70 ^bc^	2.56 ^c^	5.84 ^a^	4.79 ^ab^	0.51	0.030	0.242	0.961
Isovalerate, %	4.66	2.73	5.36	4.49	0.48	0.208	0.154	0.582
Acetate/ Propionate	3.05 ^b^	2.58 ^b^	5.32 ^a^	4.69 ^a^	0.37	0.001	0.320	0.883
**Rumen morphology**								
Rumen weight, g	639	609	499	516	30.17	0.069	0.917	0.694
Ratio to complex stomach ^1^, %	69.26	68.03	64.03	66.70	0.86	0.059	0.655	0.239
Papillae Length, μm	1137	1155	947	1254	67.96	0.743	0.255	0.308
Papillae Width, μm	105.00	93.33	96.43	120.59	4.53	0.322	0.503	0.071
Mucosal thickness, μm	44.82	36.65	42.66	44.34	1.58	0.399	0.326	0.146
Base thickness, μm	20.47	17.53	22.20	20.43	0.86	0.206	0.198	0.743

^a,b,c^ Different superscript letters in the same variable indicate statistical differences (*p* < 0.05). ^1^ The ratio of rumen weight to complex stomach weight; ^2^ HE, high energy level, ME = 10.9 MJ/Kg; LE, low energy level, ME = 8.6 MJ/Kg; HP, high protein level, CP = 15.7%; LP, low protein level, CP = 11.8%; ^3^ E = energy effect; P = protein effect; E × P = interaction effect of energy and protein; SEM = Standard error of the mean.

**Table 4 animals-10-00109-t004:** Effects of dietary energy and protein levels on the alpha diversity values of lambs.

Items	HE ^1^		LE ^1^		SEM	*p*-Value ^2^		
	HP ^1^	LP ^1^	HP	LP		E	P	E × P
OTUs	241.67 ^b^	205.33 ^b^	440.67 ^a^	410.33 ^a^	31.57	<0.01	0.064	0.852
Ace	281.33 ^b^	228.00 ^b^	483.00 ^a^	460.00 ^a^	34.07	<0.01	0.053	0.383
Chao1	294.00 ^b^	230.33 ^b^	489.68 ^a^	472.33 ^a^	34.82	<0.01	0.080	0.287
Shannon	2.67 ^b^	3.04 ^b^	4.00 ^a^	3.99 ^a^	0.19	<0.01	0.341	0.316
Simpson	0.19 ^a^	0.12 ^ab^	0.07 ^b^	0.05 ^b^	0.02	<0.01	0.088	0.232
Coverage	0.998 ^ab^	0.999 ^a^	0.997 ^c^	0.997 ^bc^	0.00	<0.01	0.468	0.855

^a,b,c^ Different superscript letters in the same variable indicate statistical differences (*p* < 0.05). ^1^ HE, high energy level, ME = 10.9 MJ/Kg; LE, low energy level, ME = 8.6 MJ/Kg; HP, high protein level, CP = 15.7%; LP, low protein level, CP = 11.8%. ^2^ E = energy effect; P = protein effect; E × P = interaction effect of energy and protein; SEM = Standard error of the mean.

**Table 5 animals-10-00109-t005:** Changes in rumen microbial community abundance from amplicon sequence data in four diets, represented as average abundance per diet.

Taxon	HE ^1^		LE ^1^		SEM	*p*-Value ^2^		
	HP ^1^	LP ^1^	HP	LP		E	P	E × P
Proteobacteria	41.90 ^a^	32.75 ^a^	2.15 ^b^	6.32 ^b^	5.61	<0.01	0.663	0.26
*p_Succinivibrionaceae_uncultured*	36.16 ^a^	28.30 ^a^	0.11 ^b^	0.47 ^b^	5.14	<0.01	0.341	0.300
Bacteroidetes	35.25 ^b^	35.59 ^b^	65.94 ^a^	41.27 ^b^	3.94	<0.01	<0.01	<0.01
*b_Bacteroidales_norank*	0.13 ^b^	0.00 ^b^	6.23 ^a^	6.03 ^a^	1.23	0.014	0.934	0.986
*b_BS11_gut_group_norank*	0.23 ^b^	0.73 ^b^	14.80 ^a^	1.73 ^b^	1.99	<0.01	<0.01	<0.01
*b_S24.7_norank*	0.07 ^c^	0.03 ^c^	0.55 ^b^	1.32 ^a^	0.17	<0.01	0.033	0.020
Firmicutes	20.98 ^b^	23.33 ^b^	25.87 ^b^	45.31 ^a^	3.62	0.029	0.063	0.129
*f_Lachnospiraceae_uncultured*	0.11 ^b^	0.30 ^b^	1.57 ^ab^	4.21 ^a^	0.61	0.012	0.127	0.178
*f_Pseudobutyrivibrio*	0.01 ^b^	0.05 ^b^	0.38 ^ab^	1.44 ^a^	0.23	0.038	0.156	0.184
*f_Ruminococcaceae_uncultured*	1.39 ^b^	1.76 ^b^	6.90 ^a^	6.51 ^a^	0.90	<0.01	0.993	0.738
*f_Ruminococcus*	1.44 ^b^	1.18 ^b^	1.39 ^b^	4.10 ^a^	0.45	0.051	0.087	0.045
*f_Saccharofermentans*	0.18 ^b^	0.00 ^b^	1.57 ^a^	0.70 ^ab^	0.23	0.014	0.157	0.332
Lentisphaerae	0.03 ^b^	0.00 ^b^	1.27 ^a^	0.91 ^ab^	0.21	<0.01	0.524	0.589
*l_RFP12_gut_group_norank*	0.03 ^b^	0.00 ^b^	1.13 ^a^	0.80 ^ab^	0.19	0.011	0.546	0.613

Only taxa abundant over 1% of the total sequences at least in one group and significantly affected by diet are presented. ^a,b,c^ Different letters in the same variable indicate statistical differences (*p* < 0.05). ^1^ HE, high energy level, ME = 10.9 MJ/Kg; LE, low energy level, ME = 8.6 MJ/Kg; HP, high protein level, CP = 15.7%; LP, low protein level, CP = 11.8%. ^2^ E = energy effect; P = protein effect; E × P = interaction effect of energy and protein. SEM = Standard error of the mean.

**Table 6 animals-10-00109-t006:** Indicator species analysis.

OTU (Genus)	Associated with	Indicator Value	*p*-Value	Relative Abundance (%)
*Succinivibrionaceae_uncultured*	HE	0.996	0.002 **	32.23
*Veillonellaceae_unclassified*	HE	0.996	0.002 **	2.90
*Veillonellaceae_uncultured*	HE	0.987	0.002 **	4.59
*Bacteroidales_norank*	LE	0.994	0.002 **	6.13
*Lachnospiraceae_uncultured*	LE	0.966	0.002 **	2.89

Associations were calculated with the Dufrene-Legendre indicator species analysis routine (Indval, indicator value) in R. Data table shows results for the analysis where rare OTUs (<1% relative abundance) were excluded. Significance levels: ** *p* ≤ 0.01. The *p*-values were corrected for multiple comparisons using the false discovery rate (FDR) with the Benjamini-Hochberg method.

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
