# Peer review of "Ruminal Microbiota and Fermentation in Response to Dietary Protein and Energy Levels in Weaned Lambs"

_animals, 2020, doi:10.3390/ani10010109_

Round 1

Reviewer 1 Report

The main objective of this paper was to evaluate the effects of dietary energy and protein levels on growth performance, microbiota and physiological properties of rumen in weaned lambs. The manuscript is an interesting paper for scientific research and people in sheep industry. However, there is no proper justification for the number of animals used in the experiment, i.e. authors should have had to perform a power test or indicate in the manuscript how the number of biological replicates was chosen. In addition, there are some grammatical and editorial errors in this manuscript needed to be well addressed.

Without statistical justification for the samples size to support why low samples sizes (n = 4) were used, I feel not enough animals were included in the treatment. L87: What size of pens? L100: Were animals allowed for ad libitum intake? Table 2 and 5: ab should be superscript. Line 111: How to collect the diet samples? Detail Line 165: QIIME version? Will raw data be made publicly available? If so, please include an accession number. Line 211: 16S Authors need to pay attention to the problem of English writing format, such as (p < 0.05). L96, L104-105, L317: there should have a space between the numerical value and unit. The values are labeled a, b, c in descending order in table 2, 3 and 5, please check table 4.

Reviewer 2 Report

Comments to the Authors:

It is well known the importance of dietary factors, including energy and protein levels, on the development of the rumen, and the rumen fermentation and microbiome. The impact of changing the level of energy and protein of the diet on these mentioned factors in lambs seems to be poorly understood. This seems to be the novelty of the article, although it is unclear and should be further emphasized. There are also other important issues to be addressed, as explained below.  

Although the experimental design looks correct, the statistical analysis should be revised. It is unclear whether the pen was included as a random effect, and this should be considered in order to try to improve the statistical analysis. Moreover, the indicator genera analysis is not explained in the material and methods section. How was it done?

In material and methods, some details should be clarified. The ADF analysis should be also explained. The rumen samples for ammonia and volatile fatty acids (VFA) are usually acidified. Did you mix your rumen samples with any acid solution? Did you use an internal standard for the VFA analysis? Please, explain clearer and justify. More details of the bioinformatics analysis could be given, such as the database used for the phylogenetic affiliation. Did you submit the sequence information for Nucleotide Sequence accession numbers?

In results, in those parameters where there is a significant interaction level of energy x level of protein (e.g., the relative proportion of Bacteroidetes), you should explain and interpret that interaction, but not the impact of each fixed effect separately.

In the discussion I recommend to express the energy levels in the same units to make easier the comparison. Another issue is the importance of the breed, do you think that some aspects of your study could be referred to this particular type of sheep? In that section, the differences among diets in the content of structural carbohydrates should be further emphasized, especially regarding the changes in the microbial community. Moreover, about this section, there is a recent article on the same subject that is only cited in the introduction  (reference number 1), although it should be cited (and several times) in the discussion, since it is similar to this study. Based on your results, do you think that a protein level of around 11% is enough for these lambs?

It might be expected that the diet energy level would affect the rumen microbial diversity, although it is unclear why the greater diversity was found with low energy diets. Why do you think this may happen? Please, discuss a bit further on this issue.

You found an increase in the relative proportion of Bacteroidetes with the low energy diets and you indicate that Bacteroidetes are able to form propionate. You stated that these results are consistent with the decrease in propionate in the low energy group, although it is the opposite. If the Bacteroidetes increased and they form propionate, you would expect an increase in propionate. Please, reword and correct.

In the discussion you mention Pseudobutyrivibrio as a dominant group in low energy diets, although that genus has not been mentioned in results in the Indicator genera analysis. Please, revise, explain and modify if necessary.

The English writing should be revised in some parts of the manuscript in order to improve the understanding.

Reviewer 3 Report

Crude protein is not going to work for this paper. Too ambiguous. This paper needs to express rumen degradable protein levels or rumen available nitrogen. 

Likewise, dietary energy levels will not work in describing energy to the rumen. The authors need to express this in terms of fermentable, carbohydrate levels supplied to the rumen.

Otherwise good paper

Author Response

Response to Reviewer 3 Comments

Point 1: Crude protein is not going to work for this paper. Too ambiguous. This paper needs to express rumen degradable protein levels or rumen available nitrogen.

 Response 1:  thanks, the author has counted the rumen degradable protein levels. Added it to table 1, and supplemented some discuss content, details see the article, please.

Point 2: Likewise, dietary energy levels will not work in describing energy to the rumen. The authors need to express this in terms of fermentable, carbohydrate levels supplied to the rumen.

 Response 2:  thanks, the author has counted the NFC levels, and ratio of NFC to NDF, ratio of RDP to ME. Added it to table 1, and supplemented some discuss content, details see the article, please.

PS: For the revised full article according to the comments of reviewer 1, 2,and 3, please see the attachment.
